# The Convergence of Polymer Science and Predictive Modeling for Noninvasive Glucose Monitoring

**DOI:** 10.3390/pharmaceutics17111488

**Published:** 2025-11-18

**Authors:** Ju-Hwan Lee, Hong-Sik Yun, Hee-Jae Jeon

**Affiliations:** 1Department of Smart Health Science and Technology, Kangwon National University, Chuncheon 24341, Republic of Korea; 202515539@kangwon.ac.kr (J.-H.L.); hongsik0523@kangwon.ac.kr (H.-S.Y.); 2Department of Mechatronics Engineering, Kangwon National University, Chuncheon 24341, Republic of Korea; 3Department of Mechanical and Biomedical Engineering, Kangwon National University, Chuncheon 24341, Republic of Korea; 4The uCare Co., Ltd., 1 Gangwondaehak-gil, Chuncheon 24341, Republic of Korea

**Keywords:** blood glucose monitoring, non-invasive sensor, molecularly imprinted polymers (MIPs), conductive polymer hydrogels (CPHs), artificial intelligence (AI), machine learning (ML), predictive modeling, wearable sensor

## Abstract

The global effort to manage diabetes effectively is driving continuous innovation in glucose monitoring devices. While current systems have improved patient care, persistent challenges with sensor stability and invasiveness highlight the need for advanced, patient-friendly technologies. A particularly promising frontier is emerging from the convergence of advanced polymer science and artificial intelligence (AI), opening new pathways for noninvasive biosensing. This feature review provides a comprehensive overview of polymer-based “hardware”, such as molecularly imprinted polymers (MIPs), conductive polymer hydrogels (CPHs), and functional coatings, which offer robust and biocompatible alternatives to traditional enzyme-based sensors. Concurrently, we examine (AI) “software”, including machine learning and predictive modeling, which enable reliable interpretation of complex biosignals for real-time glucose monitoring. Furthermore, this review highlights critical challenges in scalability, long-term in vivo stability, regulatory approval, and clinical adoption, while discussing strategies for successful translation into pharmaceutical technology and medical devices. By mapping the current landscape and future directions, this review aims to guide research toward the next generation of intelligent, patient-centric, noninvasive glucose monitoring platforms.

## 1. Introduction

Diabetes has become a global health challenge with prevalence steadily increasing worldwide, particularly in aging societies and low-resource regions [1,2]. Blood glucose control is directly influencing patients’ quality of life, as poor regulation is closely associated with complications such as cardiovascular disease, neuropathy, retinopathy, and kidney failure [3,4]. Glucose, the principal component of blood sugar, is the most abundant monosaccharide in nature and a critical energy source for essential metabolic pathways [5]. The importance of blood glucose monitoring is underscored by the biosensor market, which has grown rapidly in response to clinical demand. The market is valued at $27.4 billion, with blood glucose monitoring alone accounting for $15.34 billion [6]. This strong economic signal highlights not only the medical but also the societal and industrial urgency of developing reliable, user-friendly monitoring systems.

Despite decades of technological progress, current blood glucose monitoring technologies still face clear limitations [7,8,9]. Self-monitoring of blood glucose (SMBG) requires painful finger pricks multiple times a day and provides only single-point data, leading to poor patient compliance and preventing continuous tracking of glycemic variability [10]. Continuous glucose monitoring (CGM) revolutionized diabetes management by measuring glucose concentration in interstitial fluid (ISF) at short intervals [11]. However, physiological delays between blood and interstitial glucose, the need for frequent calibration, sensor replacement, and the inherent instability of enzyme-based electrodes remain unresolved [5,12]. These challenges indicate that, even with the latest CGM systems, the goal of truly noninvasive, accurate, and long-term glucose sensing is still unmet.

To address these shortcomings, a new paradigm is emerging that combines robust synthetic polymers with advanced AI techniques. Polymers such as molecularly imprinted polymers (MIPs), conductive polymer hydrogels (CPHs), and functional protective coatings offer superior stability, cost-effectiveness, and biocompatibility compared with enzyme-based receptors [5]. At the same time, AI enables accurate interpretation of complex, noisy biosignals, turning raw sensor outputs into clinically meaningful glucose predictions [12]. This synergy is particularly critical for noninvasive monitoring using weak and variable signals from alternative biofluids such as sweat, saliva, or breath condensate [13]. Emerging reports also demonstrate that AI-driven algorithms can forecast glucose fluctuations, offering preventive alerts rather than retrospective feedback, which represents a paradigm shift in diabetes management.

Thus, the development of next-generation glucose sensors requires coordinated progress in both materials engineering and intelligent data analytics. Rather than presenting individual advances in isolation, this review takes an integrated perspective to show how polymer-based sensing platforms and AI converge to address the persistent shortcomings of conventional technologies and to guide the design of future systems. The discussion traces the evolution of glucose monitoring methods, highlights advances in polymer chemistry, and examines the role of AI in signal interpretation and predictive modeling to demonstrate how these elements are increasingly aligned. Representative studies are included to illustrate the feasibility of combining polymer-based hardware with computational approaches, while challenges such as scalability, long-term stability, regulatory approval, and clinical acceptance are critically assessed to identify barriers to translation. Figure 1 provides a conceptual overview of this paradigm, emphasizing how synthetic materials and intelligent algorithms together establish a foundation for accurate, patient-centered, and noninvasive glucose monitoring. In addition to noninvasive electrochemical platforms, a minimally invasive/optical sensing option is shown to illustrate that implantable polymer sensors can feed the same AI/ML pipeline. An explicit data-flow arrow from the sensor to the AI/ML module has been added for clarity.

## 2. The Technological Evolution of Glucose Monitoring

Glycemic measurement technology for effective diabetes management has undergone remarkable development over the past few decades. The paradigm of diabetes management shifted with the advent of SMBG in which patients directly measure their own blood sugar, moving away from indirect methods like early urine tests [14]. However, the limitations of this invasive approach highlighted the need for more convenient and continuous data acquisition, leading to the development of CGM systems. This chapter reviews the overall development of glucose measurement technology, tracing the evolution from early intermittent methods to modern continuous systems [15].

Initial diabetes diagnosis relied on qualitative methods to check for the presence of sugar in urine. In the 20th century, chemical tests using benedict solutions were introduced, but these could only indirectly confirm blood glucose concentration [16]. The turning point came in the 1970s with the development of biochemical analysis using glucose oxidase. Based on this, the first personal blood glucose meters appeared, and the innovative concept of SMBG was born. Patients could now directly check their blood sugar levels with a trace amount of blood from their fingertips, without relying on hospital labs [17].

Throughout the 1990s and 2000s, SMBG technology rapidly evolved with a focus on miniaturization and user convenience. The required blood volume decreased to microliters, measurement times shortened to seconds, and devices began to feature memory and basic data analysis functions like “no-coding” technology and calculating blood sugar averages [18,19,20]. Since the 2010s, SMBG has transformed into a “connected health” device, with Bluetooth technology enabling automatic data transmission to smartphone apps for real-time sharing with caregivers and medical staff [21].

However, despite decades of advancement, SMBG could not overcome its fundamental limitation of providing only a single point data. It was unable to capture glucose trends or fluctuations between measurements, often missing critical events like nocturnal hypoglycemia or postprandial spikes [22]. Furthermore, the pain and discomfort from repeated fingerpricks remained a major barrier to patient compliance [23]. These shortcomings motivated the development of CGM.

By inserting a micro-sensor electrode into subcutaneous tissue, CGM systems automatically measure glucose concentration in ISF at short intervals, typically every five minutes [24]. This introduction changed the paradigm of glucose management once again. Patients and clinicians could now see not only real-time glucose levels but also their trends, rate of change, and temporal patterns, enabling proactive alerts for hypo- or hyperglycemia and more precise insulin control [25].

Commercialized in the mid-2000s, CGM technology has also seen rapid advancement. Early systems like the Medtronic guardian real-time (2006) required multiple daily calibrations [26]. Subsequent models from companies like Dexcom gradually improved accuracy and reduced this burden [27]. A major turning point was the factory-calibrated “flash” system, the Abbott freestyle libre (2014), which eliminated the need for user calibration [28]. A key development is iCGM (interoperable CGM) systems, such as the Dexcom g6 (2018), which provide continuous data streaming without calibration and serve as a core technology for hybrid closed-loop systems, or the “artificial pancreas” [29].

The evolution of CGM technology continues, with current research focused on next-generation microneedle-based sensors to further minimize pain, and the incorporation of artificial intelligence to provide personalized predictions from the vast amount of data collected [30]. In these latest systems, AI-based predictive algorithms enable early warnings for impending glycemic excursions and supporting the transition toward fully automated insulin delivery.

As such, CGM greatly improved the level of diabetes management. However, as pointed out in the introduction, it is not free from its own challenges, including the discomfort of a minimally invasive method, the physiological time lag between blood and ISF glucose, and the inherent instability of its enzyme-based sensors [31]. These fundamental barriers of even the most advanced current technology present the inevitable need for a paradigm shift from enzyme-dependent systems toward more stable synthetic polymers combined with AI-driven signal interpretation [32]. To reflect this transition, the historical evolution outlined below is followed in later sections by polymer-based sensing platforms and AI-driven glucose prediction.

## 3. Advanced Polymer Platforms for Noninvasive Sensors

The materials science underpinning next-generation sensors begins by addressing the limitations of traditional biological receptors. This section outlines the synthesis, functionality, and advantages of three representative polymer classes, which together form the hardware foundation of advanced sensing platforms. As illustrated in Figure 2, the first panel (Figure 2a) depicts MIPs, where the removal of a glucose template molecule creates nanoscale cavities that enable selective recognition. The second panel (Figure 2b) shows multilayer protective coatings, in which an outer zwitterionic layer prevents biofouling while an inner anionic layer blocks electroactive interferents, thereby improving long-term stability. The third panel (Figure 2c) presents CPHs, which combine the electrical conductivity of conjugated polymers with the flexibility and tissue-like properties of hydrogels, making them attractive substrates for wearable and implantable biosensors. Table 1 complements this overview by comparing the working mechanisms, advantages, limitations, and representative applications of these polymer platforms.

A leading example is MIPs, which are synthetic polymers featuring nanoscale cavities tailored to specific target molecules. They offer a robust alternative to unstable and expensive biological receptors like enzymes or antibodies [5]. These materials boast outstanding chemical stability, cost efficiency, and ease of fabrication [35]. The process involves polymerizing functional monomers around a template molecule and then removing it to leave complementary binding sites. Since glucose itself lacks strong binding groups, an innovative approach uses “mimetic templates” like glucuronic acid to improve imprinting toward glucose-like structures; however, the resulting selectivity generally remains lower than that of natural receptors. This limitation, together with the heterogeneity of binding sites reported for glucose MIPs, has hindered straightforward clinical translation so far [35]. Various polymerization techniques are employed for MIP synthesis. While thermal polymerization allows for mass production, it can result in poor particle uniformity. In contrast, techniques like electrospinning can generate high-surface-area nanofibers ideal for integration into wearable sweat monitors [31]. More broadly, sensors produced using this technology have also successfully detected glucose in human urine, demonstrating its wide potential for non-invasive monitoring [5].

Beyond creating these specific recognition sites, the physical structure of the sensor must be flexible and biocompatible, especially for wearable devices. This challenge is addressed by CPHs. CPHs uniquely combine the electrical activity of conductive polymers (CPs) with the biocompatibility and flexibility of hydrogels, creating an interface that mimics biological tissue while still transmitting electrical signals [34,36]. This inherent flexibility, elasticity, and self-healing capability make them an ideal substrate for wearable and implantable electrochemical biosensors (WEBSs) that must endure mechanical stress on the skin. Current research is focused on overcoming issues like limited strain detection or dehydration by forming dual network structures or introducing solvents like glycerin [36].

While these polymers provide an ideal physical platform, ensuring long-term stability within the body requires another layer of innovation. Implantable sensors face constant threats from biofouling due to protein adhesion and interference from electroactive substances such as ascorbic acid (AA) and uric acid (UA). Functional polymer coatings play a key role in addressing these issues. For example, zwitterionic polymer coatings form a strong hydration layer that repels proteins and cells, extending sensor lifespan. Meanwhile, negatively charged polymer membranes like Nafion act as protective barriers that electrostatically repel interfering substances. The most effective approach often involves a multilayer structure with an inner anti-interference layer and an outer biofouling-resistant layer, providing optimal protection while maintaining high sensitivity to glucose [33].

This overall research trend demonstrates a shift in thinking beyond simply seeking better individual materials, toward how to engineer material ‘systems’ that function robustly within complex biological environments. This is akin to assembling a complex machine from specialized components, each performing a distinct task. It suggests that materials science is moving beyond foundational discovery and into the mature engineering field of complex system integration.

## 4. AI for Signal Processing and Prediction

Advanced polymers such as MIPs and CPHs have addressed some of the stability issues inherent to enzyme-based sensors, yet the signals they generate remain complex, indirect, and often noisy. AI has a crucial role in transforming these challenging outputs into clinically meaningful information. In the following section, the focus is placed on how computational methods process raw sensor data, reduce noise, and enable reliable and predictive insights that support real-time glucose management.

The first and most fundamental task is signal processing and calibration [37]. Raw sensor signals are often contaminated by noise, baseline drift, and interference from drugs or physical activity [38]. Additionally, sensor sensitivity can change over time, a phenomenon known as sensor drift, which makes recalibration essential [39]. While early model-based techniques like Kalman filters improved accuracy over simple linear regression, machine learning (ML) models have recently proven to deliver superior performance [38]. Models such as Gaussian process regression (GPR), random forest (RF), and support vector regression (SVR) are effective for constructing complex calibration models that account for multiple variables [40,41]. Furthermore, neural network autoencoders can effectively remove noise even at very low signal-to-noise ratios [38].

Once the signal is cleaned and calibrated, the next step is to use this processed data for predictive modeling. The goal of CGM extends beyond reporting current blood glucose levels to predicting future trends, thereby preventing hypoglycemia or hyperglycemia. For this time-series forecasting task, neural network models including recurrent neural networks (RNNs) and long short-term memory (LSTM) networks. have consistently shown the highest relative performance across various prediction intervals [42]. RF has also proven to be a robust alternative, often outperforming other traditional models [12,43]. The success of these models critically depends on the large datasets generated by CGM devices, which provide abundant training data by taking measurements every one to five minutes [42].

This predictive power becomes even more critical when decoding the complex biological signals from non-invasive methods. Here, AI is not merely a useful tool but an essential element. For instance, convolutional neural networks (CNNs), which are adept at extracting spatial features, have been used to analyze spectral images of the skin with clinically excellent results [44,45,46]. Time-varying signals are ideally handled by RNNs and their variant LSTM [47]. Some cutting-edge approaches even use a dual-mode framework, combining CNNs for imaging data with RF regression for voltage readings to generate more robust predictions [46].

A crucial point is that the choice of an AI model is determined by the sensor’s physical characteristics and the data’s structure. CNNs are optimal for spatial data like images, while RNNs are suited for temporal data like CGM streams. This implies that successful polymer-AI sensor development requires more than just applying algorithms. It instead necessitates a deep expertise in selecting and tailoring artificial intelligence architecture to the specific data generated by the chosen sensing method [34].

Table 2 provides an overview of the most commonly used AI models, together with their input data, performance metrics, and practical strengths and limitations. Building on these insights, AI is fundamentally transforming diabetes management. Conventional CGM has largely offered retrospective information, whereas AI-based systems can now deliver predictive warnings of hypoglycemia within the next thirty minutes [42]. This shift enables patients to respond proactively and represents an important step toward fully automated closed-loop systems, often described as an artificial pancreas.

## 5. Representative Recent Studies of Polymer–AI Integrated Glucose Monitoring

A growing number of studies demonstrate that polymer-based sensing platforms, when combined with AI, can create synergies that transcend the limitations of conventional glucose monitoring technologies. Recent investigations have reported a range of representative approaches, including noninvasive breath analysis, implantable hydrogel-based CGM, optical spectroscopy, and paper-based point-of-care sensors. These examples collectively highlight how the integration of advanced materials with intelligent data analysis can provide both sensitivity and robustness in challenging biological environments, as summarized in Figure 3.

As shown in Figure 3a, one important development is a noninvasive breath analyzer that employs a molecularly imprinted polymer sensor to detect trace concentrations of D-glucose in exhaled breath condensate. The electrical resistance changes generated by glucose binding are processed using advanced deep learning models, including convolutional and recurrent neural networks. In particular, long short-term memory architecture has been effective in capturing the temporal dynamics of the signal. This integration, which analyzes D-glucose in exhaled breath condensate, reported a very low detection limit (0.001 ppb, a unit typically used for gaseous measurements) and a rapid response time of about 30 s, highlighting its potential for noninvasive monitoring. To enable fair comparison with other noninvasive, liquid-phase sensors (sweat, saliva, ISF), such performance is preferably expressed in molar concentration units (nM–µM) [47].

A second example, depicted in Figure 3b, involves an implantable hydrogel-based CGM platform that was fabricated using polyethylene glycol diacrylate hydrogels containing phosphorescent dyes and glucose-specific enzymes. Signal acquisition was carried out with a compact phosphorescence lifetime imager (PLI), and deep learning models were trained to perform two distinct functions. One model was responsible for assessing reader alignment, while the other classified glucose concentration levels. In vitro evaluation with a skin phantom demonstrated an overall classification accuracy of nearly eighty-nine percent across hypoglycemic, normoglycemic, and hyperglycemic ranges. Importantly, the system remained resilient under random reader misalignments of up to 4.7 mm and still maintained correct classification when misalignments exceeded 5 mm. These results indicate that the platform has strong potential to deliver stable, minimally invasive, and clinically meaningful glucose monitoring [48].

Beyond these two cases, other approaches have also expanded the scope of polymer–AI integration. Short-wave infrared spectroscopy combined with a dual-mode ML framework has shown clinically excellent performance, with convolutional neural networks analyzing spatial imaging data while random forest regression interpreted voltage signals [46]. Paper-based electrochemical sensors enhanced with a support vector regression algorithm achieved accuracies exceeding ninety-nine percent when detecting glucose in serum, demonstrating how computational techniques can overcome the resolution limitations inherent to low-cost paper devices [49]. Conventional enzyme-based electrochemical sensors have also benefited from the addition of advanced ML models. In particular, the use of eXtreme gradient boosting significantly improved predictive accuracy, yielding an R-squared value of more than ninety-two percent while simultaneously reducing error metrics compared with traditional approaches [50].

Across these diverse systems, performance has been systematically evaluated using both clinical and statistical benchmarks. The Clarke error grid continues to be regarded as the most important clinical standard, with the safe zones A and B representing the target range for acceptable medical decision-making [12]. In parallel, statistical indicators such as the root mean square error and the mean absolute error provide a quantitative measure of predictive accuracy and model robustness [51]. Collectively, the studies reviewed here emphasize that there is no universal algorithm that can be applied in every context. The optimal choice depends strongly on the physical characteristics of the sensor and the structural properties of the data it generates. Neural networks such as long short-term memory architectures and convolutional neural networks have proven highly effective in capturing temporal and spatial complexity [47,48]. In contrast, ensemble methods including support vector regression and eXtreme gradient boosting have delivered excellent results in mitigating noise, improving resolution, and providing reliable predictions from lower quality sensor signals [49,50].

A consistent lesson across all of these examples is that the use of AI enhances the inherent performance of polymer-based sensors. By compensating for motion-induced artifacts, correcting variability in measurements, and enhancing sensitivity, ML transforms raw and noisy outputs into clinically actionable information. This recurring theme underscores the future direction of glucose monitoring research, which increasingly points toward the seamless integration of advanced polymeric materials with predictive computational algorithms to provide accurate, reliable, and patient-centered solutions for diabetes management. However, most of the polymer–AI systems reviewed in this section were validated in in vitro, phantom, or small-scale settings and have not yet undergone large, real-world clinical studies.

## 6. Challenges for Clinical Translation

Advanced polymer-AI sensor systems must overcome several significant obstacles before they can achieve widespread adoption in clinical practice. These challenges can be broadly divided into two domains, namely issues related to technical maturity and issues concerning clinical application and commercialization.

From the perspective of technical maturity, several fundamental engineering limitations remain unresolved. The large-scale production of molecularly imprinted polymers remains a major bottleneck, because producing particles with uniform and reproducible binding sites is still difficult, and even solid-phase synthesis routes have not yet been demonstrated in a fully automated, industrially scalable form [52]. In addition, although protective polymer coatings have improved stability, maintaining long-term performance in vivo remains a critical hurdle, since the harsh biological environment gradually degrades sensor function over time [33,53]. Noninvasive devices add another layer of complexity, as obtaining reliable and reproducible samples from biofluids such as sweat is inherently challenging. In practice, sweat-based systems often face issues of insufficient volume and rapid evaporation [54]. Achieving the sensitivity and specificity required at clinically relevant low glucose concentrations, particularly in the hypoglycemic range, is also still problematic for many platforms [53].

The second group of challenges arises in the context of clinical application and commercialization. At the data and algorithm level, the use of AI with sensitive health information raises concerns about privacy, security, and potential bias, requiring alignment with existing data-protection regulations. Because these issues involve model transparency and population representativeness, they are discussed in more detail in the following section [55,56].

In addition to these algorithmic issues, system-level factors create further barriers. High infrastructure costs limit accessibility, particularly in low-resource settings, and the regulatory approval pathway for complex integrated systems remains both uncertain and demanding [54]. Even the most accurate sensing platform will not succeed if it is not designed with the end user in mind. Patient comfort, ease of use, and cognitive burden are critical determinants of adoption, and neglecting these human factors often leads to poor compliance [39]. Ultimately, the transition of these technologies from proof-of-concept demonstrations to reliable medical products requires not only incremental improvements under controlled laboratory conditions but also systematic strategies to overcome fundamental engineering, regulatory, and ethical barriers.

Beyond these general considerations, long-term in vivo stability is repeatedly cited as the most critical obstacle, since biofouling, fibrous encapsulation, and local inflammatory responses can drastically reduce sensor lifespan [32]. Calibration requirements, even in factory-calibrated systems, continue to impose a burden, particularly during exercise, dehydration, or fluctuating environmental conditions [32]. At the data level, the mismatch between reference blood glucose values and alternative biofluid readouts such as sweat, saliva, or breath represents a fundamental bottleneck for training robust AI algorithms. This mismatch hinders generalization across patient cohorts; specific bias-mitigation requirements for such AI models are outlined in the subsequent section [32]. While deep learning models have demonstrated superior accuracy, their clinical use depends on providing sufficient transparency for clinicians, as discussed in the next section.

Finally, clinical adoption requires more than technical success. Regulatory approval pathways for integrated AI–sensor platforms are still underdeveloped, and the absence of large-scale randomized clinical trials constitutes a major gap between proof-of-concept studies and real-world translation [32]. Moreover, without strong evidence of cost-effectiveness, patient usability, and accessibility in low-resource settings, even highly accurate technologies risk limited adoption in practice. Taken together, these challenges underscore the need for coordinated progress in engineering, clinical validation, regulatory science, and ethical governance if polymer–AI hybrid sensors are to achieve meaningful impact in diabetes care.

To make these technology-, AI-, and regulation-related constraints more explicit, the subsequent section provides a focused critical appraisal, with particular attention to MIP scalability, AI interpretability, bias, and current regulatory expectations.

## 7. Limitations and Regulatory Considerations

Section 3, Section 4 and Section 5 summarized recent advances in polymeric sensing materials and AI-based signal processing for noninvasive glucose monitoring. However, several technical, clinical, and regulatory factors still prevent these systems from reaching the level of currently deployed CGM devices. This section consolidates those constraints to clarify where polymer–AI platforms remain precommercial. MIPs have been explored as stable, low-cost synthetic receptors for glucose, but several materials-level factors still block clinical translation. Because glucose is imprinted through non-covalent interactions, the resulting polymers contain heterogeneous binding sites, which leads to batch-to-batch variability and makes sensor calibration unreliable [52,57,58,59]. Common fabrication routes such as bulk, precipitation, emulsion, or electro-polymerization improve certain aspects of processing but still require labor-intensive grinding, sieving, or template-extraction steps that are not yet scalable to industrial production [52]. Although solid-phase nanoMIP synthesis has introduced template reuse and partial automation, it has not fully resolved these scale-up constraints, and no MIP-based glucose sensor has reached the medical diagnostics market, which remains dominated by glucose oxidase-based systems [52,58].

Polymer- and MIP-based glucose sensors therefore move from materials constraints to a second layer of constraints once AI is integrated. AI-based modeling can help optimize template–monomer interactions and improve imprinting uniformity, but embedding AI into a medical sensing workflow introduces requirements for interpretability and regulatory compliance [60]. Clinical AI/ML models, especially in decision-support contexts, are now expected to provide post hoc explanations of their predictions, and complex deep or ensemble models are treated as black boxes unless methods such as SHAP or LIME are applied [61,62,63]. At the same time, medical AI is vulnerable to data and label bias, so models trained on imbalanced or non-representative datasets can perform unevenly across underrepresented subgroups, increasing the risk of healthcare inequality [61,64,65]. Recent principles published by regulators (FDA, Health Canada, MHRA) on Good machine learning practice (GMLP) emphasize transparency, documented data provenance, and demonstration of performance across the intended patient population, including age, sex, and ethnicity strata [65]. Consequently, future polymer–AI glucose monitoring systems will have to show not only sensing accuracy but also model explainability and bias mitigation across clinically relevant subgroups before they can be considered for clinical use [62,65].

For wearable or skin-interfaced implementations, power budget and on-device computation become additional design constraints, as reported for integrated and sweat-based glucose monitoring platforms [30,31,54]. Deep recurrent or dual-mode models that perform well in laboratory studies often require compression or edge-level optimization (e.g., TinyML-style drift compensation) before deployment; otherwise battery life and wireless operation are compromised [51].

Model complexity therefore has to be co-designed with device autonomy in future polymer–AI glucose platforms. From a commercialization standpoint, current enzyme-based CGM electrodes still benefit from standardized, high-volume manufacturing and low per-sensor cost, whereas most polymer- or MIP-based systems require more complex, less scalable fabrication, which at present limits their economic competitiveness [24,25,52,58]. These constraints are reflected in Table 3, which juxtaposes commercial CGM systems with the polymer–AI approaches reviewed in this article and shows that most polymer-based systems remain at in vitro, lab-scale, or early prototype maturity.

## 8. Conclusions

This review examined how advances in polymer-based sensing materials and AI can jointly push noninvasive or minimally invasive glucose monitoring beyond the constraints of enzyme-based CGM. Polymers such as MIPs, multilayer antifouling coatings, and conductive hydrogels provide the hardware basis for stable signal acquisition, while AI- and ML-based modeling enables noise reduction, calibration, and even short-term glucose prediction. At the same time, the analysis in Section 7 makes clear that materials-level issues (binding-site heterogeneity, scale-up of MIPs), AI-specific issues (interpretability, bias, subgroup performance), and device-level issues (power and edge computation for wearables) must be addressed in parallel before clinical translation can occur. Future work should therefore prioritize scalable polymer fabrication, long-term in vivo stability, and the adoption of explainable and regulation-ready AI pipelines that document data provenance and report performance across diverse patient groups. If these technical and regulatory gaps are closed, polymer–AI hybrid platforms can evolve from proof-of-concept demonstrations to clinically deployable systems and, ultimately, form part of broader patient-centered digital health infrastructures.

## Figures and Tables

**Figure 1 pharmaceutics-17-01488-f001:**
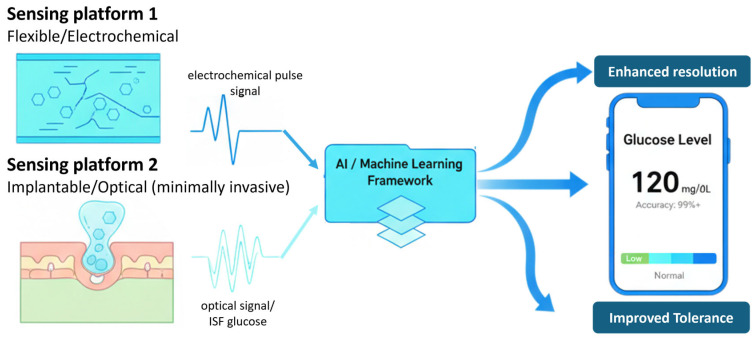
Conceptual framework of noninvasive glucose monitoring by integration polymer-based sensors with AI for signal interpretation and prediction.

**Figure 2 pharmaceutics-17-01488-f002:**
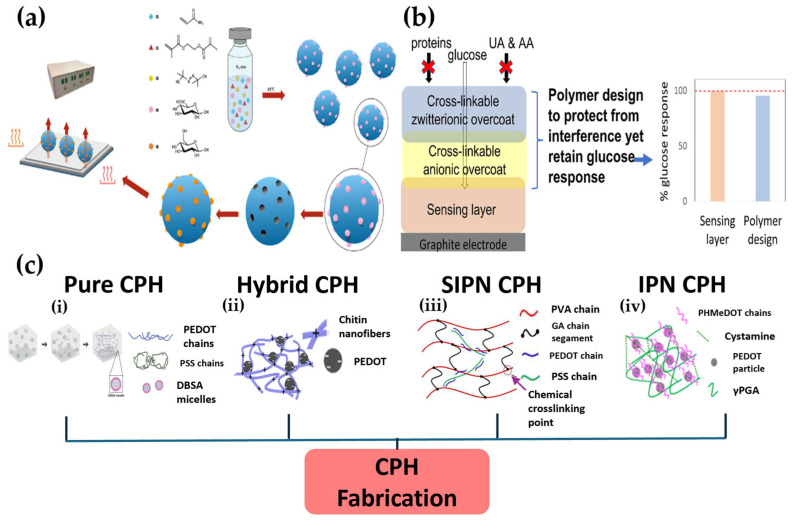
Key polymer platforms for advanced glucose sensors. This figure illustrates three key material science strategies for creating robust sensor “hardware”. (**a**) The fabrication of MIPs, where the removal of a glucose template molecule leaves a nanoscale cavity shaped for selective glucose binding. The wavy arrows represent the heat transfer (or thermal response) measured by this thermal detection-based sensor [5]. (**b**) A multilayer polymer coating designed to protect an implantable sensor, where an outer zwitterionic layer prevents biofouling from proteins and an inner anionic layer blocks interferents [33]. (**c**) The classification of CPHs based on their internal structure, categorized as (**i**) a single-component method (pure), (**ii**) a blended material method (hybrid), (**iii**) a method where a linear polymer is trapped within a network (semi-interpenetrating polymer network, SIPN), and (**iv**) a method composed of two independent, entangled networks (interpenetrating polymer network, IPN) [34].

**Figure 3 pharmaceutics-17-01488-f003:**
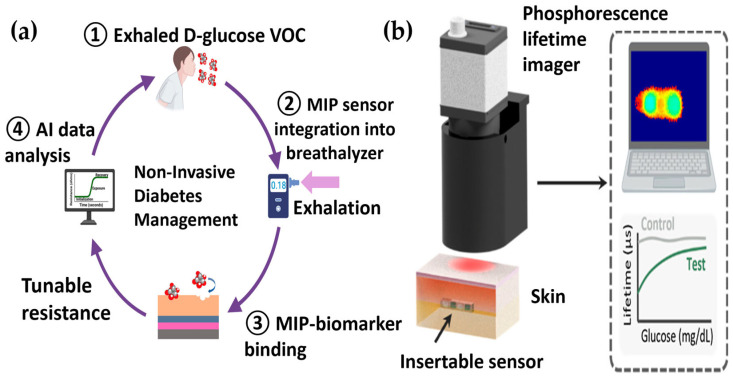
Schematics of integrated polymer-AI sensing systems. This figure illustrates two examples of integrated glucose monitoring platforms. (**a**) The operational cycle of a non-invasive system where a MIP sensor detects glucose in exhaled breath, and ML analyzes the resulting electrical signal to predict blood glucose levels [47]. (**b**) The computational analysis pipeline for an implantable, phosphorescence based CGM system, where AI processes images from an external reader to ensure reliable measurements despite motion-induced misalignment [48].

**Table 1 pharmaceutics-17-01488-t001:** Comparison of polymer platforms for blood glucose sensing.

Polymer Type	Working Mechanism	Key Advantages	Challenges	Representative Applications	Ref.
MIPs	Forms nano-cavities complementary to template molecules	High stability, low cost, a synthetic alternative to enzymes	Difficulty in mass production of uniform particles, template bleeding incomplete removal	Noninvasive sensors (urine/breath), food analysis, environmental monitoring	[5]
CPHs	Provides a tissue-like electronic interface by combining electrical activity, flexibility	High biocompatibility, Stretchability, Self-healing capability, Tissue-mimicking properties	Limited strain-sensing range, functional degradation due to swelling, signal hysteresis	Wearable and implantable sensors, flexible electronic devices, drug delivery systems	[35]
Functional protective coatings	Protects the sensor from interfering substances and biofouling via electrostatic repulsion or the formation of a hydration layer	Enhanced in vivo stability, increased selectivity, extended sensor lifespan	Potential formation of an additional diffusion barrier, issues with the long-term durability of the coating	Implantable CGM sensors, stabilization of in vivo biosensors	[34]

**Table 2 pharmaceutics-17-01488-t002:** Application of artificial models in blood glucose monitoring.

AI Model	Main Applications	Input Data Types	Key Performance Metrics	Advantages/Disadvantages	Ref.
Ensemble methods (e.g., Random forest)	Predictive modeling, calibration, classification	Tabular feature data (e.g., clinical information, sensor readings)	Accuracy, F1-Score, AUC	Robust and resistant to overfitting/ Model interpretation can be complex	[44]
Support vector machine (SVM)	Predictive modeling, classification	Feature vectors, sensor signals	Accuracy, sensitivity, precision	Effective in high-dimensional spaces/ Can be computationally intensive for large datasets	[12]
Feedforward neural networks (ANN, MLP)	Glucose level estimation, classification	Feature vectors, multi-sensor inputs	Accuracy, RMSE, MAE	Excellent for modeling non-linear relationships/ Requires large datasets and significant hyperparameter tuning	[12]
Recurrent neural networks (RNN, LSTM)	Time-series forecasting (e.g., future glucose prediction)	Continuous glucose data, temporal sensor signals	RMSE, MAE	Specialized for learning temporal patterns/ May struggle with capturing long-term dependencies	[48]
Convolutional neural networks (CNN)	Noninvasive estimation (analysis of spatial data)	Spectroscopic images, thermal images	MAPE, clarke error grid (CEG)	Highly effective for feature extraction from grid-like data/ Difficult to apply directly to sequential time-series data	[47]

**Table 3 pharmaceutics-17-01488-t003:** Comparative summary of commercial CGMs and polymer–AI-based noninvasive glucose monitoring approaches.

Category	Matrix	Performance	Clinical Relevance	Maturity	Limitation	Ref.
Commercial CGM (Dexcom G6/G7, Freestyle Libre 2/3)	ISF	Mard ≈ 9–10% (Dexcom G6); Libre Mard ≈ 9–15% (Study-dependent); Factory calibrated	Most readings in Clarke/Consensus zones A + B (>95%), acceptable for diabetes management	Commercial, large clinical use	ISF–blood time lag (~5 min); limited wear time (10–14 days); enzyme-based drift/biofouling	[24,25,28,32]
Polymer-based sensing platforms (MIPs, multilayer coatings, CPHs)	Artificial plasma, artificial sweat, urine, saliva, serum	Coatings: low relative error vs. BSA/AA/UA (≈2–3%) and 77% signal retention after 12 h; MIPs: nM–µM LOD in noninvasive biofluids; CPHs: high sensitivity in sweat and ~30-day retention	No MARD/Clarke; goal is to extend in vivo lifetime by reducing biofouling/FBR; detection shown only in noninvasive biofluids	Lab-scale, Preclinical, prototype	MIP binding-site heterogeneity; unresolved long-term in vivo stability; low/evaporative sweat volume; scale-up difficulty	[31,33,34,52,54]
SWIR/optical sensing + CNN/RF	Skin/optical	Dual-mode processing reported “Clinically excellent” performance	No MARD/Clarke; only relative accuracy in noninvasive spectral setting	Feasibility, pilot	Instrument complexity; dual-signal dependence; skin/thickness variation needs compensation	[46]
Noninvasive EBC MIP sensor + DL	Exhaled breath condensate	Reported LOD 0.001 ppb: response ≈ 30 s	No MARD/Clarke; very low EBC glucose therefore not directly comparable to commercial CGM	Proof-of-concept	EBC collection reproducibility; unit inconsistency; no simultaneous human validation	[47]
Implantable PEGDA hydrogel CGM + DL alignment	Tissue/ISF (phantom)	Three-range glucose classification ≈ 89%; DL compensates reader misalignment	Range-level only; no MARD/Clarke; not CGM-equivalent yet	In vitro, phantom prototype	Alignment dependence; long-term in vivo/immunological effects not shown; possible high-glucose saturation	[48]
Paper-based electrochemical sensor + SVR	Spiked human serum	ML accuracy > 99%; LOD ≈ 100 nM; stable response for 50 days	No MARD/Clarke; validated for spot measurements, not continuous monitoring	Lab-scale	Low-cost sensor resolution limits; no continuous real-patient validation	[49]
Enzymatic DPV sensor + XGBoost	Prepared low-concentration glucose (sweat-target)	R^2^ > 0.92 with improved MAE/RMSE	No MARD/Clarke; preparatory step toward sweat-based sensing	Model-development	No real sweat samples; incomplete compensation for practical interferences	[50]

## Data Availability

No new data were created or analyzed in this study. Data sharing is not applicable to this article.

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
