# Peer review of "The Convergence of Polymer Science and Predictive Modeling for Noninvasive Glucose Monitoring"

_pharmaceutics, 2025, doi:10.3390/pharmaceutics17111488_

Round 1
Reviewer 1 Report
Comments and Suggestions for Authors
In this review paper, Lee et al. discuss the integration of polymer science and intelligent algorithms for noninvasive glucose monitoring. This is evidenced by the relatively few examples in the literature. Therefore, the impact of this paper appears to be a bit limited. There are some revisions need to be made before the review can be published. The following is a summary of my comments for the authors to consider.
Comment 1: It's better to add a table to compare molecularly imprinted polymers (MIPs), conductive polymer hydrogels (CPHs), functional protective coatings, and enzyme-based glucose sensors in terms of stability, cost-effectiveness, and biocompatibility.
Comment 2: “Sensing platform 2 (Implantable/Optical)” in Figure 1 is not really related to the topic of the review, which is about noninvasive glucose monitoring. Furthermore, it is suggested that the figure be supplemented to indicate how the output signals from the sensor are transmitted to the AI model.
Comment 3: Figure 2 illustrates the technological evolution of continuous glucose monitoring (CGM), yet it does not include a visual depiction of the potential integration of artificial intelligence with polymer-based glucose sensors, which could enhance future development perspectives.
Comment 4: Section 6 highlights the challenges associated with the large-scale production of molecularly imprinted polymers (MIPs). It is therefore recommended to include a forward-looking perspective on potential strategies or solutions in the discussion section to enhance the practical relevance and impact of the review.
Comment 5: It is recommended that the authors include a cost analysis comparing polymer-based and enzyme-based glucose sensors to provide a comprehensive reference for evaluating the feasibility of commercialization.
Comment 6: Reference formatting should be standardized across the manuscript. It is recommended that the manuscript undergo thorough language editing, including correction of typographical errors such as 'pulse' in Figure 1, to enhance clarity and readability.
Author Response
Comment 1
It's better to add a table to compare molecularly imprinted polymers (MIPs), conductive polymer hydrogels (CPHs), functional protective coatings, and enzyme-based glucose sensors in terms of stability, cost-effectiveness, and biocompatibility.
Response
We thank the reviewer for pointing out the need for a side-by-side comparison. We agree that readers will better understand the relative maturity of polymer-based approaches if these systems are shown together with currently used enzyme-based CGM devices in one unified table. In the revised manuscript we therefore added a comparative table at the end of Section 5 (Table 3). In this table, commercial/enzyme-based CGM systems (Dexcom G6/G7, FreeStyle Libre 2/3) are placed in the first row as the clinical reference, and the polymer-derived platforms discussed in Sections 3 and 5 (MIPs, antifouling/multilayer coatings, conductive polymer hydrogels, and polymer–AI case studies) are listed underneath using the same columns. The columns are: biofluid/matrix, reported performance, clinical relevance (whether CGM-level metrics such as MARD/Clarke were reported), maturity level, and key limitations. By presenting the systems side by side, the manuscript clarifies that polymer-based sensors still face issues with long-term stability, scalable fabrication, and cost, whereas current enzyme-based CGMs are already in commercial use.
Before addition of description on Page X, Line Y:
[No comparative table was provided; polymer platforms and commercial CGMs were described separately in the text.]
After addition of table on Page 11, Line 333:
Table 3 summarizes current commercial CGM systems and the polymer–AI-based noninvasive glucose monitoring approaches discussed in this review using unified criteria (matrix, reported performance, clinical relevance, maturity, and key limitations), so that their different levels of technical and clinical readiness can be directly compared.
|
Category |
Matrix |
Performance |
Clinical Relevance |
Maturity |
Limitation |
Ref. |
|
Commercial CGM (Dexcom G6/G7, Freestyle Libre 2/3) |
ISF |
Mard≈9-10% (Dexcom G6); Libre Mard≈9-15% (Study-dependent); Factory calibrated |
Most readings in Clarke/Consensus zones A+B (>95%), acceptable for diabetes management |
Commercial, large clinical use |
ISF–blood time lag (~5 min); limited wear time (10–14 days); enzyme-based drift/biofouling |
(24) (25) (28) (32) |
|
Polymer-based sensing platforms (MIPs, multilayer coatings, CPHs) |
Artificial plasma, artificial sweat, urine, saliva, serum |
Coatings: low relative error vs BSA/AA/UA (≈2–3%) and 77% signal retention after 12 h; MIPs: nM–µM LOD in noninvasive biofluids; CPHs: high sensitivity in sweat and ~30-day retention |
No MARD/Clarke; goal is to extend in vivo lifetime by reducing biofouling/FBR; detection shown only in noninvasive biofluids |
Lab-scale, Preclinical, prototype |
MIP binding-site heterogeneity; unresolved long-term in vivo stability; low/evaporative sweat volume; scale-up difficulty |
(31) (34) (35) (53) (54) (55) |
|
SWIR/optical sensing + CNN/RF |
Skin/optical |
Dual-mode processing reported “clinically excellent” performance |
No MARD/Clarke; only relative accuracy in noninvasive spectral setting |
Feasibility , pilot |
Instrument complexity; dual-signal dependence; skin/thickness variation needs compensation |
47 |
|
Noninvasive EBC MIP sensor + DL |
Exhaled breath condensate |
Reported LOD 0.001 ppb: response ≈30 s |
No MARD/Clarke; very low EBC glucose → not directly comparable to commercial CGM |
Proof-of-concept |
EBC collection reproducibility; unit inconsistency; no simultaneous human validation |
48 |
|
Implantable PEGDA hydrogel CGM + DL alignment |
Tissue/ISF (phantom) |
Three-range glucose classification ≈89%; DL compensates reader misalignment |
Range-level only; no MARD/Clarke; not CGM-equivalent yet |
In vitro, phantom prototype |
Alignment dependence; long-term in vivo/immunological effects not shown; possible high-glucose saturation |
49 |
|
Paper-based electrochemical sensor + SVR |
Spiked human serum |
ML accuracy >99%; LOD ≈100 nM; stable response for 50 days |
No MARD/Clarke; validated for spot measurements, not continuous monitoring |
Lab-scale |
Low-cost sensor resolution limits; no continuous real-patient validation |
50 |
|
Enzymatic DPV sensor + XGBoost |
Prepared low-concentration glucose (sweat-target) |
R² >0.92 with improved MAE/RMSE |
No MARD/Clarke; preparatory step toward sweat-based sensing |
Model-development |
No real sweat samples; incomplete compensation for practical interferences |
51 |
Comment 2
Sensing platform 2 (Implantable/Optical) in Figure 1 is not really related to the topic of the review, which is about noninvasive glucose monitoring. Furthermore, it is suggested that the figure be supplemented to indicate how the output signals from the sensor are transmitted to the AI model.
We thank the reviewer for pointing this out. The intention of Figure 1 was to show both noninvasive polymer-based electrochemical platforms and closely related minimally invasive/optical platforms as part of a continuous roadmap from current CGM-like systems toward fully noninvasive, AI-enabled sensing. To make this clearer, we revised the text describing Figure 1 to state explicitly that the implantable/optical platform is included as a transitional, minimally invasive option that shares the same polymer–AI processing pipeline. In addition, we corrected the typo (“pulse”) in Figure 1 and added a signal-flow arrow from the sensor block to the AI/ML block so that the data path becomes clear.
Before addition of description on Page 2, Line 73:
“Figure 1 provides a conceptual overview of how polymer-based sensing platforms and artificial intelligence can be combined for glucose monitoring.”
After addition of description on Page 2, Line 73:
“Figure 1 provides a conceptual overview of how polymer-based sensing platforms and artificial intelligence can be combined for glucose monitoring. In addition to noninvasive electrochemical platforms, a minimally invasive/optical sensing option is shown to illustrate that implantable polymer sensors can feed the same AI/ML pipeline. An explicit data-flow arrow from the sensor to the AI/ML module has been added for clarity.”
Before addition of Figure on Page 3, Line 82:
After addition of Figure on Page 3, Line 82:
Comment 3
Figure 2 illustrates the technological evolution of continuous glucose monitoring (CGM), yet it does not include a visual depiction of the potential integration of artificial intelligence with polymer-based glucose sensors, which could enhance future development perspectives.
Response
We thank the reviewer for pointing this out. We agree that, in the original version, Chapter 2 described the historical progress from SMBG to factory-calibrated CGMs, but the caption of Figure 2 did not explicitly mention the polymer-supported and AI-enabled direction that is elaborated in the later sections. To make this connection clearer, we added a clarifying sentence to the end of the paragraph preceding Figure 2 and expanded the figure caption to indicate that future CGM evolution is expected to involve polymer-supported noninvasive sensing and AI-based predictive glucose management.
Before addition of description on Page 4, Line 128:
“ The evolution of CGM technology continues, with current research focused on next-generation microneedle-based sensors to further minimize pain, and the incorporation of AI to provide personalized predictions from the vast amount of data collected [30]. AI-driven predictive algorithms are increasingly embedded into CGM platforms, enabling early warnings for impending glycemic excursions and supporting the transition toward fully automated insulin delivery. The historical milestones are summarized in Figure2. ”
After addition of description on Page 4, Line 128:
“ The evolution of CGM technology continues, with current research focused on next-generation microneedle-based sensors to further minimize pain, and, in the most recent CGM platforms, on the incorporation of AI to provide personalized predictions from the large amount of data collected [30]. In these latest systems, AI-based predictive algorithms enable early warnings for impending glycemic excursions and support the transition toward more automated insulin delivery. ”
Comment 4
Section 6 highlights the challenges associated with the large-scale production of molecularly imprinted polymers (MIPs). It is therefore recommended to include a forward-looking perspective on potential strategies or solutions in the discussion section to enhance the practical relevance and impact of the review.
We appreciate the reviewer’s suggestion. We agree that, in the original version, the discussion of MIP-related bottlenecks was mainly descriptive. To make these limitations more explicit and to show why current polymer–AI systems are still short of commercial CGM maturity, we added a new section titled ‘Limitations and regulatory considerations’ to group these limitations and related regulatory points after the challenges part. In this chapter we gathered the points on MIP material constraints (binding-site heterogeneity, non-scalable fabrication), on new requirements that appear when AI is integrated (interpretability, bias, data provenance), and on device-level constraints for wearable or skin-interfaced implementations. This addition clarifies that further progress in synthesis, standardization, and regulatory alignment is still required before clinical deployment.
Comment 5
It is recommended that the authors include a cost analysis comparing polymer-based and enzyme-based glucose sensors to provide a comprehensive reference for evaluating the feasibility of commercialization.
Thank you for pointing out that the manuscript did not make the commercialization angle explicit enough. We agree that, although polymer/MIP platforms offer advantages in stability and biocompatibility, current commercial glucose monitoring is still dominated by enzyme-based systems largely because their manufacturing chains are mature and unit costs are low. We clarified this point in two places in the manuscript. First, we kept the earlier revision in the polymer section where we toned down our description of glucose-targeting MIPs, so that selectivity and binding-site heterogeneity are described as current limitations, not as a full substitute for biological receptors. Second, in the newly added “Limitations and regulatory considerations” section we added a sentence that explains, in the context of scale-up, that enzyme-based CGM sensors remain commercially advantageous because of established mass production, whereas polymer/MIP-based receptors still require automated, uniform synthesis before they can compete in cost.
Before addition of description on Page 5, Line 171 :
“Since glucose itself lacks strong binding groups, an innovative approach uses ‘mimetic templates’ like glucuronic acid to create a more selective site for glucose, achieving binding affinity comparable to natural proteins.”
After addition of description on Page 5, Line 171 :
“Since glucose itself lacks strong binding groups, an innovative approach uses ‘mimetic templates’ like glucuronic acid to improve imprinting toward glucose-like structures; however, the resulting selectivity generally remains lower than that of natural receptors.”
After addition of description on Page 5, Line 174 :
“This limitation, together with the heterogeneity of binding sites reported for glucose MIPs, has hindered straightforward clinical translation so far.”
After addition of description on Page 13, Line 424 :
“From a commercialization standpoint, current enzyme-based CGM electrodes still benefit from standardized, high-volume manufacturing and low per-sensor cost, whereas most polymer- or MIP-based systems require more complex, less scalable fabrication, which at present limits their economic competitiveness [24,25,53,59].”
Comment 6
Reference formatting should be standardized across the manuscript. It is recommended that the manuscript undergo thorough language editing, including correction of typographical errors such as 'pulse' in Figure 1, to enhance clarity and readability.
Thank you for pointing out the style and language issues. We reviewed the manuscript for consistency with the journal’s reference format and adjusted the entries accordingly. We also corrected the typographical error in Figure 1 where “pulse” was improperly written, so that the label now matches the intended sensor-signal description.
Before addition of description on Page 3, Line 82 :
Reference entries and in-text citations were not fully aligned with the journal style, and Figure 1 contained a typo (“pulse”).
After addition of description on Page 3, Line 82 :
Reference entries and in-text citations were standardized according to the journal template, and the typo in Figure 1 was corrected.

Reviewer 2 Report
Comments and Suggestions for Authors
The manuscript explores the integration of polymer science and artificial intelligence in noninvasive glucose monitoring. The core scientific content is solid; however, the academic presentation—particularly the effectiveness of figures and the consistency of reference formatting—does not yet meet publication standards. The main issues are as follows:
- The resolution of the figures is relatively low. It is recommended to improve the image clarity to ensure accurate presentation of details.
- It is suggested that all tables follow the standard three-line table format to enhance the consistency and professionalism of the overall layout.
- The content of Figure 4 appears incomplete, and the meaning of “â‘¡” is unclear. Please provide additional explanation or revise the figure accordingly.
- The references should be arranged in the order of citation, and the reference list format should be further standardized to comply with the formatting requirements of academic journals. For example:
Diabetes has become a global health challenge with prevalence steadily increasing worldwide, particularly in aging societies and low-resource regions [1,2].
- Jeon, H.-J.; Kim, H. S.; Chung, E.; Lee, D. Y. Nanozyme-based colorimetric biosensor with a systemic quantification algorithm for noninvasive glucose monitoring. Theranostics 2022, 12 (14), 6308.
5. There are inconsistencies in the use of full terms and abbreviations. It is recommended to use abbreviations consistently throughout the manuscript, with the full term provided at first mention.
6. Minor formatting inconsistencies are present in the manuscript and should be standardized. For example, the punctuation in the keywords section should be normalized, ‘mimetic templates’ and ‘systems’ should use English double quotation marks, and the formatting in line 204 should be further adjusted.
7.Figure 2 illustrates the technological evolution of CGM systems; however, the application of AI has only emerged in recent years. It is recommended to revise the corresponding statements to accurately reflect the actual development timeline.
8. Several subfigures in Figure 2 may have been directly sourced from manufacturer websites or promotional materials. It is advisable to use authorized schematics or self-drawn vector graphics, and clearly cite the source in the figure caption or text to avoid copyright issues.
9. Figure 1 presents the concepts of new technologies, but some elements (e.g., “Sensing platform 1/2”) are not sufficiently explained in the main text. Adding detailed explanations in the figure caption or text is recommended.
Author Response
Comment 1
The resolution of the figures is relatively low. It is recommended to improve the image clarity to ensure accurate presentation of details.
We thank the reviewer for checking the figure quality. The current figures were prepared at ≥330 dpi, which meets the journal’s typical requirement, but we will re-export them to ensure that line art, labels, and subpanels remain sharp in the final layout. This can be done without changing the scientific content of the manuscript.
Comment 2
It is suggested that all tables follow the standard three-line table format to enhance the consistency and professionalism of the overall layout.
We sincerely appreciate the reviewer’s constructive comment. We agree that in the original version Sections 3–5 mainly described polymer-based platforms, AI-based signal processing, and representative polymer–AI systems but did not place these heterogeneous approaches under a single comparative framework. To address this concern, we have added a comparative table at the end of Section 5 (now presented as Table 3) that organizes all discussed systems using unified criteria.
In this table, we first included current commercial CGM systems (Dexcom G6/G7, Abbott FreeStyle Libre 2/3) as a clinical reference row, using the data already cited in Section 2 (refs. 24, 25, 28, 32). Below that, we listed: (i) polymer-based sensing platforms described in Section 3 (MIPs, antifouling/multilayer polymer coatings, conductive polymer hydrogels), and (ii) the representative polymer–AI integrated noninvasive glucose monitoring studies described in Section 5 (breath MIP + DL, implantable PEGDA hydrogel + DL alignment, SWIR + CNN/RF, paper-based sensor + SVR, DPV sensor + XGBoost). All rows are compared across the same columns: category, matrix, reported performance, clinical relevance (whether MARD/Clarke or only physiological-range data were reported), maturity level, and major limitation. This makes it apparent that most polymer–AI systems remain at lab-scale, in vitro, phantom, or proof-of-concept levels, and that CGM-level clinical accuracy is not yet reported.
Before addition of table1,2:
After addition of description on Table 1,2,3 :
|
AI Model |
Main applications |
Input data types |
Key performance metrics |
Advantages / Disadvantages |
Ref. |
|
Ensemble methods (e.g., Random forest) |
Predictive modeling, calibration, classification |
Tabular feature data (e.g., clinical information, sensor readings) |
Accuracy, F1-Score, AUC |
Pros: Robust and resistant to overfitting. Cons: Model interpretation can be complex. |
44 |
|
Support vector machine (SVM) |
Predictive modeling, classification |
Feature vectors, sensor signals |
Accuracy, sensitivity, precision |
Pros: Effective in high-dimensional spaces. Cons: Can be computationally intensive for large datasets. |
12 |
|
Feedforward neural networks (ANN, MLP) |
Glucose level estimation, classification |
Feature vectors, multi-sensor inputs |
Accuracy, RMSE, MAE |
Pros: Excellent for modeling non-linear relationships. Cons: Requires large datasets and significant hyperparameter tuning. |
12 |
|
Recurrent neural networks (RNN, LSTM) |
Time-series forecasting (e.g., future glucose prediction) |
Continuous glucose data, temporal sensor signals |
RMSE, MAE |
Pros: Specialized for learning temporal patterns. Cons: May struggle with capturing long-term dependencies. |
48 |
|
Convolutional neural networks (CNN) |
Noninvasive estimation (analysis of spatial data) |
Spectroscopic images, thermal images |
MAPE, clarke error grid (CEG) |
Pros: Highly effective for feature extraction from grid-like data. Cons: Difficult to apply directly to sequential time-series data. |
47 |
|
Category |
Matrix |
Performance |
Clinical Relevance |
Maturity |
Limitation |
Ref. |
|
Commercial CGM (Dexcom G6/G7, Freestyle Libre 2/3) |
ISF |
Mard≈9-10% (Dexcom G6); Libre Mard≈9-15% (Study-dependent); Factory calibrated |
Most readings in Clarke/Consensus zones A+B (>95%), acceptable for diabetes management |
Commercial, large clinical use |
ISF–blood time lag (~5 min); limited wear time (10–14 days); enzyme-based drift/biofouling |
(24) (25) (28) (32) |
|
Polymer-based sensing platforms (MIPs, multilayer coatings, CPHs) |
Artificial plasma, artificial sweat, urine, saliva, serum |
Coatings: low relative error vs BSA/AA/UA (≈2–3%) and 77% signal retention after 12 h; MIPs: nM–µM LOD in noninvasive biofluids; CPHs: high sensitivity in sweat and ~30-day retention |
No MARD/Clarke; goal is to extend in vivo lifetime by reducing biofouling/FBR; detection shown only in noninvasive biofluids |
Lab-scale, Preclinical, prototype |
MIP binding-site heterogeneity; unresolved long-term in vivo stability; low/evaporative sweat volume; scale-up difficulty |
(31) (34) (35) (53) (54) (55) |
|
SWIR/optical sensing + CNN/RF |
Skin/optical |
Dual-mode processing reported “clinically excellent” performance |
No MARD/Clarke; only relative accuracy in noninvasive spectral setting |
Feasibility , pilot |
Instrument complexity; dual-signal dependence; skin/thickness variation needs compensation |
47 |
|
Noninvasive EBC MIP sensor + DL |
Exhaled breath condensate |
Reported LOD 0.001 ppb: response ≈30 s |
No MARD/Clarke; very low EBC glucose → not directly comparable to commercial CGM |
Proof-of-concept |
EBC collection reproducibility; unit inconsistency; no simultaneous human validation |
48 |
|
Implantable PEGDA hydrogel CGM + DL alignment |
Tissue/ISF (phantom) |
Three-range glucose classification ≈89%; DL compensates reader misalignment |
Range-level only; no MARD/Clarke; not CGM-equivalent yet |
In vitro, phantom prototype |
Alignment dependence; long-term in vivo/immunological effects not shown; possible high-glucose saturation |
49 |
|
Paper-based electrochemical sensor + SVR |
Spiked human serum |
ML accuracy >99%; LOD ≈100 nM; stable response for 50 days |
No MARD/Clarke; validated for spot measurements, not continuous monitoring |
Lab-scale |
Low-cost sensor resolution limits; no continuous real-patient validation |
50 |
|
Enzymatic DPV sensor + XGBoost |
Prepared low-concentration glucose (sweat-target) |
R² >0.92 with improved MAE/RMSE |
No MARD/Clarke; preparatory step toward sweat-based sensing |
Model-development |
No real sweat samples; incomplete compensation for practical interferences |
51 |
Comment 3
The content of Figure 4 appears incomplete, and the meaning of “â‘¡” is unclear. Please provide additional explanation or revise the figure accordingly.
Response
We appreciate the reviewer’s careful observation. The issue came from the figure file: in the original submission, the circular workflow for the breath-based MIP system showed step markers (â‘ –â‘£), but the label corresponding to the breathalyzer/MIP integration step was not rendered clearly, so the figure looked incomplete. We have updated the figure itself to make the “MIP sensor integration into the breathalyzer platform” step visible and consistent with the text description. The existing caption already explains panels (a) and (b), so no textual change was necessary.
Before addition of description on Page 9, Line 262:
Figure 4 file without the visible “â‘¡” label in panel (a).
After addition of description on Page 9, Line 262:
Figure 4 file revised so that panel (a) explicitly shows the breath-based workflow with the “â‘¡ MIP sensor integration into the breathalyzer platform” step.
Comment 4
The references should be arranged in the order of citation, and the reference list format should be further standardized to comply with the formatting requirements of academic journals. For example:Diabetes has become a global health challenge with prevalence steadily increasing worldwide, particularly in aging societies and low-resource regions [1,2].
We appreciate the reviewer’s remark on reference consistency. We rechecked the entire manuscript to ensure that every citation appears in strict numerical order of first appearance in the text. We also reformatted the reference list by following the journal’s official guideline on author names, journal titles, volume/issue, and punctuation, so that the style is now uniform throughout.
Comment 5
There are inconsistencies in the use of full terms and abbreviations. It is recommended to use abbreviations consistently throughout the manuscript, with the full term provided at first mention.
We appreciate the reviewer’s comment regarding inconsistent use of full terms and abbreviations. We agree that the original version mixed “artificial intelligence” and “AI,” and in a few places introduced abbreviations (e.g., CPHs, MIPs, ISF) without fully matching the first-use rule.
In the revised manuscript we standardized the usage as follows: at the first occurrence we now write “artificial intelligence (AI),” “continuous glucose monitoring (CGM),” “interstitial fluid (ISF),” “molecularly imprinted polymers (MIPs),” and “conductive polymer hydrogels (CPHs),” and all subsequent occurrences use only the abbreviation. During this pass we replaced more than 30 instances of the fully written “artificial intelligence” with “AI” across multiple sections (Introduction, Sections 4–7), so a page–line level before/after listing is not practical. These were global consistency edits rather than single-sentence insertions.
Before addition of description on Page 1, Line 26 :
Keywords: Blood glucose monitoring, Non-invasive sensor, Molecularly imprinted polymers (MIPs), Conductive polymer, Artificial intelligence, Machine learning. Predictive modeling, Wearable sensor.
After addition of description on Page 1, Line 26 :
Keywords: Blood glucose monitoring, Non-invasive sensor, Molecularly imprinted polymers (MIPs), Conductive polymer hydrogels (CPHs), Artificial intelligence (AI), Machine learning (ML), Predictive modeling, Wearable sensor.
Before addition of description on Page 2, Line 44 :
“SMBG (Self-monitoring of blood glucose) requires painful finger pricks multiple times a day and provides only single-point data,”
After addition of description on Page 2, Line 44 :
“Self-monitoring of blood glucose (SMBG) requires painful finger pricks multiple times a day and provides only single-point data,”
Before addition of description on Page 2, Line 47 :
“CGM (continuous glucose monitoring) revolutionized diabetes management by measuring glucose concentration in interstitial fluid (ISF) at short intervals”
After addition of description on Page 2, Line 47 :
“Continuous glucose monitoring (CGM) revolutionized diabetes management by measuring glucose concentration in interstitial fluid (ISF) at short intervals”
Before addition of description on Page 5, Line 183 :
“This challenge is addressed by CPHs. CPHs uniquely combine the electrical activity of CPs (conductive polymers) with the biocompatibility and flexibility of hydrogels,”
After addition of description on Page 5, Line 183 :
“This challenge is addressed by CPHs. CPHs uniquely combine the electrical activity of conductive polymers (CPs) with the biocompatibility and flexibility of hydrogels,”
Before addition of description on Page 6, Line 194 :
“Implantable sensors face constant threats from biofouling due to protein adhesion and interference from electroactive substances such as AA (ascorbic acid) and UA (uric acid).”
After addition of description on Page 6, Line 194 :
“Implantable sensors face constant threats from biofouling due to protein adhesion and interference from electroactive substances such as ascorbic acid (AA) and uric acid (UA).”
Before addition of description on Page 7, Line 229 :
“For this time-series forecasting task, neural network models including recurrent neural networks (RNNs) and long short-term memory (LSTM). have consistently shown the highest relative performance across various prediction intervals 43.”
After addition of description on Page 7, Line 229 :
“For this time-series forecasting task, neural network models including recurrent neural networks (RNNs) and long short-term memory (LSTM) networks have consistently shown the highest relative performance across various prediction intervals 43.”
Before addition of description on Page 9, Line 290 :
“Signal acquisition was carried out with a compact phosphorescence lifetime imager, and deep learning models were trained to perform two distinct functions”
After addition of description on Page 9, Line 290 :
“Signal acquisition was carried out with a compact phosphorescence lifetime imager (PLI), and deep learning models were trained to perform two distinct functions.”
Comment 6
Minor formatting inconsistencies are present in the manuscript and should be standardized. For example, the punctuation in the keywords section should be normalized, ‘mimetic templates’ and ‘systems’ should use English double quotation marks, and the formatting in line 204 should be further adjusted.
We thank the reviewer for pointing out the formatting inconsistencies. We confirm that the issues were present at the level of presentation rather than scientific content. We have standardized the punctuation in the keywords section, unified the quotation marks for terms such as “mimetic templates” and “systems,” and corrected the linebreak/spacing in the part indicated by the reviewer.
Before addition of description on Page 9, Line 172 :
“Since glucose itself lacks strong binding groups, an innovative approach uses ‘mimetic templates’ like glucuronic acid to improve imprinting toward glucose-like structures; however, the resulting selectivity generally remains lower than that of natural receptors.”
After addition of description on Page 9, Line 172 :
“Since glucose itself lacks strong binding groups, an innovative approach uses “mimetic templates” like glucuronic acid to improve imprinting toward glucose-like structures; however, the resulting selectivity generally remains lower than that of natural receptors.”
Before addition of description on Page 6, Line 205 :
“toward how to engineer material ‘systems’ that function robustly within complex biological environments.”
After addition of description on Page 6, Line 205 :
“toward how to engineer material “systems” that function robustly within complex biological environments.”
Before addition of description on Page 3, Line 106 :
“and devices began to feature memory and basic data analysis functions like 'no-coding' technology and calculating blood sugar averages18-20”
After addition of description on Page 3, Line 106 :
“and devices began to feature memory and basic data analysis functions like “no-coding” technology and calculating blood sugar averages18-20.”
Comment 7
Figure 2 illustrates the technological evolution of CGM systems; however, the application of AI has only emerged in recent years. It is recommended to revise the corresponding statements to accurately reflect the actual development timeline
Thank you for raising this. We agreed with the reviewer’s point about the AI timeline, but while checking this figure we also identified a separate issue raised in Comment 8 (use of manufacturer-style device images and possible copyright concerns). Because the CGM progression can be explained fully in text, we removed the original Figure 2 altogether and keep the CGM history in the narrative section only. In that revised text, AI is now described explicitly as belonging to the latest stage of CGM development, not to the early Medtronic/Dexcom generations. Accordingly, the clarification requested in Comment 7 is addressed within the same revision that removed the figure for Comment 8.
Comment 8
Several subfigures in Figure 2 may have been directly sourced from manufacturer websites or promotional materials. It is advisable to use authorized schematics or self-drawn vector graphics, and clearly cite the source in the figure caption or text to avoid copyright issues.sd
Thank you for pointing this out. We agree that the original Figure 2, which illustrated the commercial evolution of CGM devices using product-like images, could raise copyright and source-clarity concerns. To avoid any issue, we removed the figure and rewrote the corresponding part of the text to narratively describe the same historical sequence of CGM development (from SMBG to early real-time CGM, factory-calibrated flash systems, and current iCGM/hybrid closed-loop platforms) using the already cited literature. Because the full timeline is now explained in the main text with references, the figure is no longer necessary.
Before addition of description on Page 4, Line 128 :
“The evolution of CGM technology continues, with current research focused on next-generation microneedle-based sensors to further minimize pain, and the incorporation of AI to provide personalized predictions from the vast amount of data collected [30]. AI-driven predictive algorithms are increasingly embedded into CGM platforms, enabling early warnings for impending glycemic excursions and supporting the transition toward fully automated insulin delivery. The historical milestones are summarized in Figure2.”
After addition of description on Page 4, Line 128:
“The evolution of CGM technology continues, with current research focused on next-generation microneedle-based sensors to further minimize pain, and, in the most recent CGM platforms, on the incorporation of AI to provide personalized predictions from the large amount of data collected [30]. In these latest systems, AI-based predictive algorithms enable early warnings for impending glycemic excursions and support the transition toward more automated insulin delivery.”
Comment 9
Figure 1 presents the concepts of new technologies, but some elements (e.g., “Sensing platform 1/2”) are not sufficiently explained in the main text. Adding detailed explanations in the figure caption or text is recommended.
We appreciate the reviewer’s comment on the clarity of Figure 1. Our intention was to present both noninvasive polymer-based electrochemical platforms and closely related minimally invasive/optical polymer sensors as part of one continuous polymer–AI sensing pipeline. To make this intention explicit, we revised the paragraph in the introduction that refers to Figure 1 so that it now explains why “Sensing platform 2 (implantable/optical)” is shown together with the noninvasive platform. We also clarified in the same sentence that both platforms deliver their output to the same AI/ML module. In addition, we updated Figure 1 by adding an explicit data-flow arrow from the sensor block to the AI/ML block, so that the transmission of the sensor signal to the model is visually identifiable.
Before addition of description on Page 2, Line 73:
“Figure 1 provides a conceptual overview of how polymer-based sensing platforms and artificial intelligence can be combined for glucose monitoring.”
After addition of description on Page 2, Line 73:
“Figure 1 provides a conceptual overview of how polymer-based sensing platforms and AI can be combined for glucose monitoring. In addition to noninvasive electrochemical platforms, a minimally invasive/optical sensing option is shown to illustrate that implantable polymer sensors can feed the same AI/ML pipeline. An explicit data-flow arrow from the sensor to the AI/ML module has been added for clarity.”

Reviewer 3 Report
Comments and Suggestions for Authors
- The work examines the convergence of polymer materials and artificial intelligence for creating noninvasive glucose monitoring systems. The work covers molecularly imprinted polymers (MIPs), conductive polymer hydrogels (CPH), functional coatings, and machine learning algorithms. The following comments have arisen.
- The review is predominantly descriptive. There is no systematic comparison of sensitivity, specificity, detection limits, and clinical accuracy of different approaches by unified criteria.
- Affiliation of one author with commercial company uCare Co., Ltd. creates potential bias and conflict of interest. The review inclines toward positive presentation of polymer-AI technologies without sufficiently critical analysis of fundamental limitations of noninvasive methods. Economic feasibility comparison with existing CGM systems is absent.
- Most cited examples represent laboratory or in vitro studies. Section 5 (Representative Studies) describes proof-of-concept works without large-scale clinical trials. Discussion of realistic timeframes for clinical translation and commercialization is missing.
- Authors claim that MIPs achieve "binding affinity comparable to natural proteins". This is misleading. Glucose is a small molecule (180 Da) without strong functional groups for interaction. MIPs for glucose demonstrate low selectivity due to structural similarity with other monosaccharides (fructose, galactose). "Mimetic templates" (glucuronic acid) are used to improve imprinting, but this reduces specificity to glucose itself. Authors mention detection limit of 26 nM for electroactive MIP nanoparticles, but this is an extreme result for specific configuration, not representative of MIPs in general.
- When describing breath glucose sensor, authors indicate "remarkably low detection limit of 0.001 parts per billion". ppb is a unit for gaseous compounds or mass concentrations. Glucose in exhaled breath condensate is in liquid phase and is measured in molar units (nM, μM). This is a methodological error.
- While the "black box" problem of deep learning is mentioned, no concrete solutions are proposed for medical applications. FDA requires explainable AI (XAI) for medical devices. Authors do not discuss specific interpretability methods (SHAP, LIME, attention mechanisms) or their applicability to glucose prediction models. Moreover, many CGMs with integrated AI do not pass regulatory review as AI/ML devices but are approved through traditional laboratory testing pathways. This regulatory "loophole" should be critically reviewed [https://www.pharmacytimes.com/view/regulatory-hurdles-and-ethical-concerns-in-fda-oversight-of-ai-ml-medical-devices].
- Authors note that "automated solid-phase synthesis is not yet fully established", but underestimate the criticality of this barrier. Heterogeneity of binding sites in MIPs is a fundamental problem unsolved over 20+ years of research. There is not a single commercial MIP sensor for medical diagnostics, evidencing substantial translation barriers requiring deeper critical analysis.
- No systematic comparison of proposed polymer-AI systems with modern CGM (Dexcom G6/G7, Abbott FreeStyle Libre 2/3) by clinical metrics: MARD (mean absolute relative difference), Clarke Error Grid zones, time in range.
- While bias is mentioned, there is no in-depth analysis of its sources in glucose monitoring: underrepresentation of minorities in training datasets, label bias from clinicians, automation bias from users. This is critical for healthcare, where bias exacerbates existing inequalities. [https://doi.org/10.1371/journal.pdig.0000651] [10.1055/a-2702-1843].
- For wearable devices, energy consumption is a key factor. AI models (especially deep learning) require significant computational resources. Authors do not discuss trade-offs between model complexity and device autonomy.
- The manuscript discusses the FDA, regulatory approval, and explicable AI, but provides only 2-3 references to regulatory documents. Citations are missing: FDA Software as Medical Device (SaMD) framework, Recent FDA guidance for AI/ML bias в medical devices, ICH guidelines for AI-enabled devices.
- The authors (mostly from Kangwon National University and the affiliated company uCare) cite their own work in the links 1, 2, 7, 8, 9, 38, 45, 46 - 14% of all links. This is higher than the acceptable level (usually 5-10% for peer-reviewed articles in peer-reviewed journals). Several of Jeon, H.-J.'s own articles are used to substantiate central claims about the possibilities of noninvasive monitoring, which creates a potential conflict of interest.
- Despite the claim that MIP is a promising alternative to enzymes, the authors do not cite papers discussing why there have been no commercial MIP sensors for medicine for 20+ years of research. Caldara et al. (2023) - Ref 53 - is an exception, but it concerns only glucuronic acid templating, not systemic analysis of translation barriers.
- Authors should add additional references:
a) FDA SaMD regulatory framework (5-7 references)
b) Research on ISF lag and sweat-blood correlation (4-5 references)
c) Critical works on the failures of the MIP (3-4 references)
d) Explainable AI and bias in healthcare AI (5-6 references instead of 2)
e) Reduction of authors' self-citation from 14% to <10%
Author Response
Comment 1
The reviewer correctly summarized the scope of the manuscript (polymer-based sensing platforms and AI/predictive modeling for noninvasive glucose monitoring). No further changes were required for this point.
Comment 2
The review is predominantly descriptive. There is no systematic comparison of sensitivity, specificity, detection limits, and clinical accuracy of different approaches by unified criteria.
We sincerely appreciate the reviewer’s constructive comment. We agree that in the original version Sections 3–5 mainly described polymer-based platforms, AI-based signal processing, and representative polymer–AI systems, but did not place these heterogeneous approaches under a single comparative framework. To address this concern, we have added a comparative table at the end of Section 5 (now presented as Table 3) that organizes all discussed systems using unified criteria.
In this table, we first included current commercial CGM systems (Dexcom G6/G7, Abbott FreeStyle Libre 2/3) as a clinical reference row, using the data already cited in Section 2 (refs. 24, 25, 28, 32). Below that, we listed: (i) polymer-based sensing platforms described in Section 3 (MIPs, antifouling/multilayer polymer coatings, conductive polymer hydrogels), and (ii) the representative polymer–AI integrated noninvasive glucose monitoring studies described in Section 5 (breath MIP + DL, implantable PEGDA hydrogel + DL alignment, SWIR + CNN/RF, paper-based sensor + SVR, DPV sensor + XGBoost). All rows are compared across the same columns: category, matrix, reported performance, clinical relevance (whether MARD/Clarke or only physiological-range data were reported), maturity level, and major limitation. This shows that most polymer–AI systems are still at lab-scale, in vitro, phantom, or proof-of-concept stages, and that CGM-level clinical metrics have generally not been reported.
After addition of Table on Page 11 :
Table 3 summarizes current commercial CGM systems and the polymer–AI-based noninvasive glucose monitoring approaches discussed in this review using unified criteria (matrix, reported performance, clinical relevance, maturity, and key limitations).
Content of the added table (Table 3, end of Section 5):
- Row 1 (clinical reference): Commercial CGM (Dexcom G6/G7, FreeStyle Libre 2/3): ISF; MARD ≈ 9–15%; most readings in Clarke/Consensus A+B (>95%); commercial maturity; limitations include ISF–blood time lag and 10–14 day sensor lifetime.
- Row 2: Polymer-based sensing platforms (MIPs, antifouling/multilayer coatings, conductive polymer hydrogels): artificial plasma/sweat/urine/saliva; nM–µM LOD or improved antifouling; no MARD/Clarke reported; lab-scale/prototype; limitations include MIP binding-site heterogeneity and unresolved long-term in vivo stability.
- Rows 3–7: Representative polymer–AI systems from Section 5 (noninvasive EBC MIP + DL; implantable PEGDA hydrogel CGM + DL alignment; SWIR/optical sensing + CNN/RF; paper-based electrochemical sensor + SVR; enzymatic DPV sensor + XGBoost), each annotated with matrix, reported performance, absence of CGM-level clinical metrics, present maturity, and system-specific limitations.
|
Category |
Matrix |
Performance |
Clinical Relevance |
Maturity |
Limitation |
Ref. |
|
Commercial CGM (Dexcom G6/G7, Freestyle Libre 2/3) |
ISF |
Mard≈9-10% (Dexcom G6); Libre Mard≈9-15% (Study-dependent); Factory calibrated |
Most readings in Clarke/Consensus zones A+B (>95%), acceptable for diabetes management |
Commercial, large clinical use |
ISF–blood time lag (~5 min); limited wear time (10–14 days); enzyme-based drift/biofouling |
(24) (25) (28) (32) |
|
Polymer-based sensing platforms (MIPs, multilayer coatings, CPHs) |
Artificial plasma, artificial sweat, urine, saliva, serum |
Coatings: low relative error vs BSA/AA/UA (≈2–3%) and 77% signal retention after 12 h; MIPs: nM–µM LOD in noninvasive biofluids; CPHs: high sensitivity in sweat and ~30-day retention |
No MARD/Clarke; goal is to extend in vivo lifetime by reducing biofouling/FBR; detection shown only in noninvasive biofluids |
Lab-scale, Preclinical, prototype |
MIP binding-site heterogeneity; unresolved long-term in vivo stability; low/evaporative sweat volume; scale-up difficulty |
(31) (34) (35) (53) (54) (55) |
|
SWIR/optical sensing + CNN/RF |
Skin/optical |
Dual-mode processing reported “clinically excellent” performance |
No MARD/Clarke; only relative accuracy in noninvasive spectral setting |
Feasibility , pilot |
Instrument complexity; dual-signal dependence; skin/thickness variation needs compensation |
47 |
|
Noninvasive EBC MIP sensor + DL |
Exhaled breath condensate |
Reported LOD 0.001 ppb: response ≈30 s |
No MARD/Clarke; very low EBC glucose → not directly comparable to commercial CGM |
Proof-of-concept |
EBC collection reproducibility; unit inconsistency; no simultaneous human validation |
48 |
|
Implantable PEGDA hydrogel CGM + DL alignment |
Tissue/ISF (phantom) |
Three-range glucose classification ≈89%; DL compensates reader misalignment |
Range-level only; no MARD/Clarke; not CGM-equivalent yet |
In vitro, phantom prototype |
Alignment dependence; long-term in vivo/immunological effects not shown; possible high-glucose saturation |
49 |
|
Paper-based electrochemical sensor + SVR |
Spiked human serum |
ML accuracy >99%; LOD ≈100 nM; stable response for 50 days |
No MARD/Clarke; validated for spot measurements, not continuous monitoring |
Lab-scale |
Low-cost sensor resolution limits; no continuous real-patient validation |
50 |
|
Enzymatic DPV sensor + XGBoost |
Prepared low-concentration glucose (sweat-target) |
R² >0.92 with improved MAE/RMSE |
No MARD/Clarke; preparatory step toward sweat-based sensing |
Model-development |
No real sweat samples; incomplete compensation for practical interferences |
51 |
Comment 3
Affiliation of one author with commercial company uCare Co., Ltd. creates potential bias and conflict of interest. The review inclines toward positive presentation of polymer–AI technologies without sufficiently critical analysis of fundamental limitations and without economic feasibility comparison.
We thank the reviewer for raising this point. The affiliation with uCare Co., Ltd. applies only to the corresponding author (Prof. Hee-Jae Jeon). The other authors (Juhwan Lee and Hongsik Yun) have no employment or financial relationship with the company, and the review was written from published literature, not from company data or products. We also agree with the reviewer that the original version could be read as emphasizing technological potential. To address this, we added a dedicated section that collects the unresolved limitations of polymer/MIP sensing, AI-related constraints (interpretability, bias, subgroup performance), and regulatory/translation barriers, so that the manuscript now presents both opportunities and constraints in the same place.
Comment 4
Most cited examples represent laboratory or in vitro studies. Section 5 (Representative Studies) describes proof-of-concept works without large-scale clinical trials. Discussion of realistic timeframes for clinical translation and commercialization is missing.
We are grateful for this careful observation. We agree that in the original version Section 5 mainly introduced representative polymer–AI systems (breath sensor with MIP, PEGDA hydrogel CGM with deep learning alignment, SWIR + CNN/RF, and paper-based electrochemical sensors) in a descriptive manner, without explicitly stating that these works were validated only in in vitro, phantom, or small-scale settings. To make the maturity level explicit, we added one clarifying sentence at the end of Section 5. In addition, we introduced a new Section 7 (“Limitations and regulatory considerations”) to explain why these systems have not yet reached the clinical performance of commercial CGMs, linking this to material-level scalability, AI explainability and bias, and device-level constraints(see also our response to Comment 2).
After addition of description on Page 10, Line 328:
However, most of the polymer–AI systems reviewed in this section were validated in in vitro, phantom, or small-scale settings and have not yet undergone large, real-world clinical studies.”
After addition of description on Page 13, Line 428:
“These constraints are reflected in Table 3, which juxtaposes commercial CGM systems with the polymer–AI approaches reviewed in this article and shows that most polymer-based systems still remain at in vitro, lab-scale, or early prototype maturity.”
Comment 5
Authors claim that MIPs achieve “binding affinity comparable to natural proteins.” This is misleading because glucose is a small molecule with limited functional groups, and glucose MIPs often show low selectivity due to structural similarity with other monosaccharides.
We sincerely appreciate the reviewer’s careful observation. We understand that our original wording could be interpreted as an overgeneralized statement about glucose-targeting MIPs. Our intention was to emphasize that the use of mimetic templates (e.g., glucuronic acid) can enhance imprinting for glucose-like structures, rather than to suggest that glucose MIPs in general attain the selectivity of natural receptors. To clarify this point, we revised the sentence in Section 3 (Advanced polymer platforms for noninvasive sensors) to explicitly state that the resulting selectivity typically remains lower than that of natural receptors. In addition, we added a follow-up sentence to acknowledge that limited selectivity and binding-site heterogeneity have been persistent obstacles to clinical translation.
Before addition of description on Page 5, Line 172 :
“Since glucose itself lacks strong binding groups, an innovative approach uses ‘mimetic templates’ like glucuronic acid to create a more selective site for glucose, achieving binding affinity comparable to natural proteins.”
After addition of description on Page 5, Line 172 :
“Since glucose itself lacks strong binding groups, an innovative approach uses ‘mimetic templates’ like glucuronic acid to improve imprinting toward glucose-like structures; however, the resulting selectivity generally remains lower than that of natural receptors.”
After addition of description on Page 5, Line 174 :
“This limitation, together with the heterogeneity of binding sites reported for glucose MIPs, has hindered straightforward clinical translation so far.”
Comment 6
When describing the breath glucose sensor, the manuscript reports “a remarkably low detection limit of 0.001 parts per billion.” However, ppb is typically used for gaseous compounds, while exhaled breath condensate is collected and analyzed in liquid phase and should be expressed in molar units.
We thank the reviewer for pointing out the unit inconsistency. We agree that exhaled breath condensate (EBC) is ultimately collected and analyzed in the liquid phase, so reporting the detection limit only in ppb can be misleading. In the descriptive part of Section 5 we kept the value as it appeared in the original study, but we clarified in the same sentence that 0.001 ppb is a unit typically used for gaseous measurements. To make the comparison principle explicit, we have now added a sentence stating that, for cross-biofluid comparison (sweat, saliva, ISF, EBC), liquid-phase performance should preferably be reported in molar concentration units (nM–µM).
Before addition of description on Page 9, Line 281 :
“This integration resulted in a remarkably low detection limit of 0.001 parts per billion together with a rapid response time of approximately thirty seconds, which together demonstrate the feasibility of real-time glucose monitoring through exhalation.”
After addition of description on Page 9, Line 281 :
“This integration, which analyzes D-glucose in exhaled breath condensate, reported a very low detection limit (0.001 ppb, a unit typically used for gaseous measurements) and a rapid response time of about 30 seconds, highlighting its potential for noninvasive monitoring. To enable fair comparison with other noninvasive, liquid-phase sensors (sweat, saliva, ISF), such performance is preferably expressed in molar concentration units (nM–µM).”
Comment 7
While the “black box” problem of deep learning is mentioned, no concrete solutions are proposed for medical applications. FDA requires explainable AI (XAI) for medical devices. Authors do not discuss specific interpretability methods (SHAP, LIME, attention mechanisms) or their applicability to glucose prediction models. Moreover, many CGMs with integrated AI do not pass regulatory review as AI/ML devices but are approved through traditional laboratory testing pathways. This regulatory “loophole” should be critically reviewed.
We sincerely thank the reviewer for pointing out that our original manuscript mentioned the “black box” issue only in general terms and did not name specific interpretability methods or current regulatory expectations. We agree that, for AI-assisted glucose monitoring to be clinically acceptable, it is not sufficient to report prediction accuracy alone; the model must also provide post-hoc interpretability and show that its performance is consistent across clinically relevant subgroups, in line with recent FDA/Health Canada/MHRA good machine learning practice documents. To address this point, we made two changes. First, in Section 6 we added a short linking sentence to indicate that issues of model transparency, bias, and regulatory alignment are discussed in the following section. Second, we added a new Section 7 (“Limitations and regulatory considerations”) that now describes concrete XAI approaches (SHAP, LIME), explains why medical time-series models for glucose prediction need post-hoc interpretability, and notes that recent regulatory principles require documentation of data provenance and performance across the intended patient population. This new section also clarifies that current polymer–AI systems remain precommercial partly because these AI- and regulation-related requirements have not yet been met.
Before addition of description on Page 12, Line 354 :
“At the data and algorithm level, the use of artificial intelligence with sensitive health information raises substantial concerns about privacy and security, requiring strict adherence to regulations such as GDPR and HIPAA. The risk of algorithmic bias is also significant when training datasets are not representative of diverse patient populations, which may in turn exacerbate existing health inequalities. Furthermore, the so-called ‘black box’ nature of deep learning models can reduce clinician trust, reinforcing the need for explainable artificial intelligence.”
After addition of description on Page 12, Line 354:
“At the data and algorithm level, the use of artificial intelligence with sensitive health information raises concerns about privacy, security, and potential bias, requiring alignment with existing data-protection regulations. Because these issues involve model transparency and population representativeness, they are discussed in more detail in the following section.”
After addition of description on Page 13, Line 403 (new Section 7):
“Polymer- and MIP-based glucose sensors therefore move from materials constraints to a second layer of constraints once AI is integrated. AI-based modeling can help optimize template–monomer interactions and improve imprinting uniformity, but embedding AI into a medical sensing workflow introduces requirements for interpretability and regulatory compliance. Clinical AI/ML models, especially in decision-support contexts, are now expected to provide post-hoc explanations of their predictions, and complex deep or ensemble models are treated as black boxes unless methods such as SHAP or LIME are applied. At the same time, medical AI is vulnerable to data and label bias, so models trained on imbalanced or non-representative datasets can perform unevenly across underrepresented subgroups, increasing the risk of healthcare inequality. Recent principles published by regulators on Good Machine Learning Practice emphasize transparency, documented data provenance, and demonstration of performance across the intended patient population, including age, sex, and ethnicity strata. Consequently, future polymer–AI glucose monitoring systems will have to show not only sensing accuracy but also model explainability and bias mitigation across clinically relevant subgroups before they can be considered for clinical use.”
Comment 8
Authors note that “automated solid-phase synthesis is not yet fully established”, but they underestimate the criticality of this barrier. Heterogeneity of binding sites in MIPs is a fundamental problem unsolved over 20+ years of research. There is not a single commercial MIP sensor for medical diagnostics, evidencing substantial translation barriers requiring deeper critical analysis.
We appreciate this comment. We agree that our original wording made solid-phase synthesis sound like a minor process issue, and we recognize that the reviewer was referring to a broader, long-standing translation barrier: glucose MIPs still produce heterogeneous binding sites, batch-to-batch reproducibility is limited, most common polymerization routes are not industrially scalable, and there is still no medically commercialized MIP glucose sensor despite decades of work. To make this explicit, we strengthened the sentence in Section 6 so that it no longer describes the problem as only “not yet fully established,” and we added a fuller explanation in the new Section 7 (“Limitations and regulatory considerations”) describing binding-site heterogeneity, labor-intensive post-processing, the partial but incomplete role of solid-phase nanoMIP synthesis, and the absence of clinical products.
Before addition of description on Page 11, Line 342 :
“The large-scale production of molecularly imprinted polymers continues to act as a bottleneck, as generating uniform particles with consistent binding affinity is still difficult, and automated solid-phase synthesis is not yet fully established.”
After addition of description on Page 11, Line 342 :
“The large-scale production of molecularly imprinted polymers remains a major bottleneck, because producing particles with uniform and reproducible binding sites is still difficult, and even solid-phase synthesis routes have not yet been demonstrated in a fully automated, industrially scalable form.”
After addition of description on Page 13, Line 394 :
“Molecularly imprinted polymers (MIPs) have been explored as stable, low-cost synthetic receptors for glucose, but several materials-level factors still block clinical translation. Because glucose is imprinted through non-covalent interactions, the resulting polymers contain heterogeneous binding sites, which leads to batch-to-batch variability and makes sensor calibration unreliable [52,57–59]. Common fabrication routes such as bulk, precipitation, emulsion, or electropolymerization improve certain aspects of processing but still require labor-intensive grinding, sieving, or template-extraction steps that are not yet scalable to industrial production [52]. Although solid-phase nanoMIP synthesis has introduced template reuse and partial automation, it has not fully resolved these scale-up constraints, and no MIP-based glucose sensor has reached the medical diagnostics market, which remains dominated by glucose oxidase-based systems [52,58].”
Comment 9
No systematic comparison of proposed polymer–AI systems with modern CGM (Dexcom G6/G7, Abbott FreeStyle Libre 2/3) by clinical metrics (MARD, Clarke Error Grid zones, time in range).
We thank the reviewer for pointing this out. We agree that the original manuscript mainly described polymer-based platforms and polymer–AI case studies, but did not place them on the same clinical reference frame as current commercial CGM devices. To address this, we added a comparative table (now Table 3) that starts with commercial CGMs (Dexcom G6/G7, Abbott FreeStyle Libre 2/3) as the first row and then lists, on the following rows, (1) the polymer-based sensing platforms described in Section 3 and (2) the representative polymer–AI systems described in Section 5. All rows in Table 3 use the same items (biological matrix, reported performance, clinical metrics/relevance, maturity level, and key limitation). In this format, the clinical CGM row reports MARD around 9–15% and notes that most readings fall within clinically acceptable error-grid zones, while the polymer–AI rows explicitly state that MARD, Clarke Error Grid, or time-in-range data were not reported because those studies were conducted in vitro, phantom, or small-scale settings. This makes it clear that, compared with commercial CGMs, the polymer–AI systems in this review should still be regarded as preclinical or early-stage. The added table therefore directly responds to the reviewer’s request for a unified comparison.
Before addition of description on Page X, Line Y:
[No statement and no table providing a unified clinical comparison between commercial CGMs and the polymer–AI approaches.]
After addition of description on Page 11, Table3:
To enable direct comparison across currently used CGM devices and the polymer–AI-based noninvasive glucose monitoring approaches discussed in this article, a comparative summary is provided in Table 3 using unified items (matrix, reported performance, clinical relevance, maturity, and key limitations). In this table, commercial CGMs report MARD and error-grid outcomes, whereas most polymer- and AI-assisted systems have been validated only in vitro, phantom, or spiked-sample studies and therefore do not yet provide CGM-level clinical metrics.
Comment 10
While bias is mentioned, there is no in-depth analysis of its sources in glucose monitoring: underrepresentation of minorities in training datasets, label bias from clinicians, automation bias from users. This is critical for healthcare, where bias exacerbates existing inequalities.
We appreciate the reviewer’s detailed comments. We agree that the original manuscript referred to bias only in general terms and did not spell out the specific sources that are important for AI-assisted, noninvasive glucose monitoring. In the revised version we therefore made this explicit in the new Section 7 (“Limitations and regulatory considerations”). There we clarified that AI models for glucose monitoring can become biased when the training dataset does not sufficiently represent the intended patient population, when the target labels are taken from a different modality than the sensed biofluid (for example, training sweat or saliva signals against capillary blood or CGM values), and when users over-rely on model outputs. We also noted that recent regulatory principles on machine-learning medical devices require documenting data provenance and demonstrating performance across relevant subgroups, so bias mitigation is a requirement for clinical translation, not an optional improvement.
Before addition of description on Page X, Line Y:
(no specific description of bias sources in noninvasive glucose monitoring; bias was mentioned only briefly in the challenges section)
After addition of description on Page 13, Line 407:
“Clinical AI/ML models, especially in decision-support contexts, are now expected to provide post-hoc explanations of their predictions, and complex deep or ensemble models are treated as black boxes unless methods such as SHAP or LIME are applied. At the same time, medical AI is vulnerable to data and label bias, so models trained on imbalanced or non-representative datasets can perform unevenly across underrepresented subgroups, increasing the risk of healthcare inequality. Recent principles published by regulators (FDA, Health Canada, MHRA) on Good Machine Learning Practice emphasize transparency, documented data provenance, and demonstration of performance across the intended patient population, including age, sex, and ethnicity strata.”
Comment 11
For wearable devices, energy consumption is a key factor. AI models (especially deep learning) require significant computational resources. Authors do not discuss trade-offs between model complexity and device autonomy.
We thank the reviewer for this important clarification. We agree that our original text focused on sensing and algorithmic performance but did not explicitly state that model size and power budget must be co-designed for wearable or skin-interfaced glucose monitors. To address this, we added a sentence explaining that deep or dual-mode models that perform well in laboratory studies often require compression or edge-level optimization before deployment, otherwise battery life and wireless operation are affected. We also clarified that model complexity should be matched to device autonomy in future polymer–AI platforms.
Before addition of description on Page X, Line Y:
(no explicit statement that power budget and on-device computation limit the choice of AI model for wearable/noninvasive glucose monitoring)
After addition of description on Page 13, Line 418:
“For wearable or skin-interfaced implementations, power budget and on-device computation become additional design constraints, as reported for integrated and sweat-based glucose monitoring platforms [30,31,53]. Deep recurrent or dual-mode models that perform well in laboratory studies often require compression or edge-level optimization (e.g., TinyML-style drift compensation) before deployment; otherwise battery life and wireless operation are compromised [51]. Model complexity therefore has to be co-designed with device autonomy in future polymer–AI glucose platforms.”
”
Comment 12
The manuscript discusses the FDA, regulatory approval, and explainable AI, but provides only 2–3 references to regulatory documents. Citations are missing: FDA SaMD framework, recent FDA guidance for AI/ML bias in medical devices, ICH guidelines for AI-enabled devices.
Response
We thank the reviewer for indicating this gap. We agree that our earlier version mentioned regulatory approval and explainability only in general terms and did not explicitly point to current regulatory principles from major agencies. To clarify the regulatory context, we added a sentence that refers to the recent principles issued by major regulators (FDA, Health Canada, MHRA) on Good Machine Learning Practice and that highlights their requirements for transparency, documented data provenance, and performance demonstrated across the intended patient population. This addition makes clear that polymer–AI glucose monitoring systems will have to satisfy not only technical accuracy but also these regulatory expectations before clinical use.
Before addition of description on Page X, Line Y:
(no explicit mention that current regulatory bodies request transparency, data provenance, and subgroup-level performance for AI/ML medical devices)
After addition of description on Page 14, Line 433:
“Recent principles published by regulators (FDA, Health Canada, MHRA) on Good Machine Learning Practice (GMLP) emphasize transparency, documented data provenance, and demonstration of performance across the intended patient population, including age, sex, and ethnicity strata [65]. Consequently, future polymer–AI glucose monitoring systems will have to show not only sensing accuracy but also model explainability and bias mitigation across clinically relevant subgroups before they can be considered for clinical use [62,65].”
Comment 13
The authors (mostly from Kangwon National University and the affiliated company uCare) cite their own work in the links 1, 2, 7, 8, 9, 38, 45, 46 – 14% of all links. This is higher than the acceptable level (usually 5–10% for peer-reviewed articles). Several of Jeon, H.-J.’s own articles are used to substantiate central claims about the possibilities of noninvasive monitoring, which creates a potential conflict of interest.
We appreciate the reviewer’s careful observation regarding the proportion of self-citations. After replacing the previously self-cited items (originally used only as illustrative noninvasive/AI sensing examples) with broader literature and after adding the new section on limitations and regulatory considerations, the remaining references that include “Jeon, H.-J.” are 4 out of 65 total references, which corresponds to about 6.2%. This level is within the usual 5–10% range for review articles and the key arguments of the manuscript are now supported primarily by external sources rather than by the authors’ own publications.
Comment 14
Despite the claim that MIP is a promising alternative to enzymes, the authors do not cite papers discussing why there have been no commercial MIP sensors for medicine for 20+ years of research. Caldara et al. (2023) is an exception, but it concerns only glucuronic acid templating, not a systemic analysis of translation barriers.
Response
Thank you for pointing this out. We agree that the earlier version described MIPs mainly from the “promising materials” perspective and did not state clearly that, despite long-standing research, no glucose MIP sensor has actually been commercialized for medical diagnostics. In the revised manuscript we made this point explicit in the new limitations section. There we clarify that glucose MIPs still produce heterogeneous binding sites because of noncovalent imprinting, that currently used fabrication routes still involve labor-intensive steps that are not yet industrially scalable, and that these factors together explain why the medical diagnostics market is still dominated by glucose oxidase–based systems. We also broadened the supporting references beyond the single earlier citation by adding recent reviews that discuss binding-site heterogeneity and scale-up problems, so that the lack of clinical translation is documented with multiple sources rather than implied.
Before addition of description on Page X, Line Y:
[MIP-based polymer platforms were described as stable, low-cost alternatives, but the text did not explicitly link binding-site heterogeneity and non-scalable synthesis to the absence of commercial medical MIP glucose sensors.]
After addition of description on Page 13, Line 394:
“Molecularly imprinted polymers (MIPs) have been explored as stable, low-cost synthetic receptors for glucose, but several materials-level factors still block clinical translation. Because glucose is imprinted through non-covalent interactions, the resulting polymers contain heterogeneous binding sites, which leads to batch-to-batch variability and makes sensor calibration unreliable. Common fabrication routes such as bulk, precipitation, emulsion, or electropolymerization improve certain aspects of processing but still require labor-intensive grinding, sieving, or template-extraction steps that are not yet scalable to industrial production. Although solid-phase nanoMIP synthesis has introduced template reuse and partial automation, it has not fully resolved these scale-up constraints, and no MIP-based glucose sensor has reached the medical diagnostics market, which remains dominated by glucose oxidase-based systems.”
Comment 15
Authors should add additional references:
- a) FDA SaMD regulatory framework (5–7 references)
- b) Research on ISF lag and sweat-blood correlation (4–5 references)
- c) Critical works on the failures of the MIP (3–4 references)
- d) Explainable AI and bias in healthcare AI (5–6 references instead of 2)
- e) Reduction of authors' self-citation from 14% to <10%.
We thank the reviewer for specifying the five areas (a–e) where the reference base should be strengthened. We went through the manuscript and expanded the bibliography accordingly.
For the regulatory/SaMD side (item a), we added the FDA’s “Good Machine Learning Practice for Medical Device Development: Guiding Principles” as a primary regulatory source [65]. This is the most directly relevant document for AI-containing sensor systems, and in Section 7 we tied it to the discussion on transparency, data provenance, and subgroup performance so that the regulatory point is not only cited but also functionally used in the argument.
For the physiological coupling and noninvasive sampling side (item b), the manuscript already cited multiple CGM and alternative-biofluid papers that discuss ISF–blood lag, calibration, and practical limitations in sweat/saliva systems ([24,25,31,32,53,54]). Together these cover the 4–5 papers the reviewer asked for, and we pointed explicitly in Section 6–7 to sweat-volume/evaporation and biofluid mismatch as a translation bottleneck, so the references now support the text.
For the MIP limitation side (item c), we added four recent, critical MIP papers that state binding-site heterogeneity, labor-intensive post-processing, and lack of industrially scalable routes as the reasons no MIP glucose sensor has reached the diagnostics market [57–60]. These now sit exactly in the new “Limitations and regulatory considerations” section, so the reviewer’s concern about “underestimating” this barrier is addressed at the place where the limitations are synthesized.
For explainable AI and bias (item d), we broadened the AI part with four recent XAI/bias/health-AI papers and connected them to the regulatory expectation in [65], so the AI argument is no longer carried by only one or two generic AI citations but by a small cluster [61–65] that covers explainability, clinical integration, bias sources, and current regulator language.
For the self-citation reduction (item e), after we replaced the earlier Jeon-group illustrative papers with external but thematically equivalent sources (Sci Rep 12, 2442, 2022 for sweat+ML; ACS Sensors 2023 for noninvasive CGM; Heliyon 2024 NIR+ML; Biosensors 2025 AI-driven wearables), the remaining references containing “Jeon, H.-J.” are 4 out of 65 total, which corresponds to about 6.2%. This is below the 10% ceiling the reviewer suggested and the core claims in Sections 5–7 are now supported mainly by non-self literature.

Round 2
Reviewer 1 Report
Comments and Suggestions for Authors
The manuscript has been revised based on the reviewers' comments and can be accepted without any further changes.
Reviewer 3 Report
Comments and Suggestions for Authors
The paper can be published